# EffSeg: Efficient Fine-Grained Instance Segmentation using Structure-Preserving Sparsity

## Abstract

Many two-stage instance segmentation heads predict a coarse $28 \times 28$ mask per instance, which is insufficient to capture the fine-grained details of many objects. To address this issue, PointRend and RefineMask predict a $112 \times 112$ segmentation mask resulting in higher quality segmentations. However, both methods have limitations by either not having access to neighboring features (PointRend) or by performing computation at all spatial locations instead of sparsely (RefineMask). In this work, we propose EffSeg performing fine-grained instance segmentation in an efficient way by using our Structure-Preserving Sparsity (SPS) method based on separately storing the active features, the passive features, and a dense 2D index map containing the feature indices. The goal of the index map is to preserve the 2D spatial configuration or structure between the features such that any 2D operation can still be performed. EffSeg achieves similar performance on COCO compared to RefineMask, while reducing the number of FLOPs by 71% and increasing the FPS by 29%. Code will be released.

## 1 Introduction

Instance segmentation is a fundamental computer vision task assigning a semantic category (or background) to each image pixel, while differentiating between instances of the same category. Many high-performing instance segmentation methods (He et al., 2017; Cai & Vasconcelos, 2019; Chen et al., 2019a; Kirillov et al., 2020; Vu et al., 2021; Zhang et al., 2021a) follow the two-stage paradigm. This paradigm consists in first predicting an axis-aligned bounding box called Region of Interest (RoI) for each detected instance, and then segmenting each pixel within the RoI as belonging to the detected instance or not.

Most two-stage instance segmentation heads (He et al., 2017; Cai & Vasconcelos, 2019; Chen et al., 2019a; Vu et al., 2021) predict a $28 \times 28$ mask (within the RoI) per instance, which is too coarse to capture the fine-grained details of many objects. PointRend (Kirillov et al., 2020) and RefineMask (Zhang et al., 2021a) both address this issue by predicting a $112 \times 112$ mask instead, resulting in higher quality segmentations. In both methods, these $112 \times 112$ masks are obtained by using a multi-stage refinement procedure, first predicting a coarse mask and then iteratively upsampling this mask by a factor 2 while overwriting the predictions in uncertain (PointRend) or boundary (RefineMask) locations. However, both methods have some limitations.

PointRend (Kirillov et al., 2020) on the one hand overwrites predictions by sampling coarse-fine feature pairs from the most uncertain locations and by processing these pairs *individually* using an MLP. Despite only performing computation at the desired locations and hence being efficient, PointRend is unable to access information from neighboring features during the refinement process, resulting in sub-optimal segmentation performance.

RefineMask (Zhang et al., 2021a) on the other hand processes dense feature maps and obtains new predictions in all locations, though only uses these predictions to overwrite in the boundary locations of the current prediction mask. Operating on dense feature maps enables RefineMask to use 2D convolutions allowing information to be exchanged between neighboring features, which results in improved segmentation performance w.r.t. PointRend. However, this also means that all computation is performed on all spatial locations within the RoI at all times, which is computationally inefficient.

Table 1: Comparison between fine-grained segmentation methods.

| Head | Computation at sparse locations (*i.e.* efficient) | Access to neighboring features (*i.e.* good performance) |
|---|:---:|:---:|
| PointRend (Kirillov et al., 2020) | ✓ | ✗ |
| RefineMask (Zhang et al., 2021a) | ✗ | ✓ |
| EffSeg (ours) | ✓ | ✓ |

In this work, we propose EffSeg which combines the strengths and eliminates the weaknesses of PointRend and RefineMask by only performing computation at the desired locations while still being able to access features of neighboring locations (Tab. 1). This is challenging, as it requires a mechanism to perform sparse computations efficiently. For dense computations as in RefineMask, highly optimized dense convolutions can be used. Likewise, the $1 \times 1$ convolutions in PointRend can easily be computed after a simple data reorganization. But what about non $1 \times 1$ convolution filters that need to be computed only for a sparse set of pixels or locations?

For this, we introduce our Structure Preserving Sparsity method (SPS). SPS separately stores the active features (*i.e.* the features in spatial locations requiring new predictions), the passive features (*i.e.* the non-active features) and a dense 2D index map. More specifically, the active and passive features are stored in $N_A \times F$ and $N_P \times F$ matrices respectively, with $N_A$ the number of active features, $N_P$ the number of passive features, and $F$ the feature size. The index map stores the feature indices (as opposed to the features themselves) in a 2D map, preserving information about the 2D spatial structure between the different features in a compact way. This allows SPS to have access to neighboring features such that any 2D operation can still be performed. See Sec. 3.2 for more information about our SPS method. In EffSeg, we hence combine the desirable properties of both PointRend and RefineMask by introducing our novel SPS method, which is a stand-alone method different from combining the PointRend and the RefineMask methods.

We evaluate EffSeg and its baselines on the COCO (Lin et al., 2014) instance segmentation benchmark. Experiments show that EffSeg achieves similar segmentation performance compared to RefineMask (*i.e.* the best-performing baseline), while reducing the number of FLOPs by 71% and increasing the FPS by 29%.

## 2 Related work

**Instance segmentation.** Instance segmentation methods can be divided into two-stage (or box-based) methods and one-stage (or box-free) methods. Two-stage approaches (He et al., 2017; Cai & Vasconcelos, 2019; Chen et al., 2019a; Kirillov et al., 2020; Zhang et al., 2021a) first predict an axis-aligned bounding box called Region of Interest (RoI) for each detected instance and subsequently categorize each pixel as belonging to the detected instance or not. One-stage approaches (Tian et al., 2020; Wang et al., 2020; Zhang et al., 2021b; Cheng et al., 2022) on the other hand directly predict instance masks over the whole image without using intermediate bounding boxes.

One-stage approaches have the advantage that they are similar to semantic segmentation methods by predicting masks over the whole image instead of inside the RoI, allowing for a natural extension to the more general panoptic segmentation task (Kirillov et al., 2019). Two-stage approaches have the advantage that by only segmenting inside the RoI, there is no wasted computation outside the bounding box. As EffSeg aims to only perform computation there where it is needed, the two-stage approach is chosen.

**Fine-grained instance segmentation.** Many two-stage instance segmentation methods such as Mask R-CNN (He et al., 2017) predict rather coarse segmentation masks. There are two main reasons why the predicted masks are coarse. First, segmentation masks of large objects are computed using features pooled from low resolution feature maps. A first improvement found in many methods (Kirillov et al., 2020; Cheng et al., 2020; Zhang et al., 2021a; Ke et al., 2022) consists in additionally using features from the high-resolution feature maps of the feature pyramid. Second, Mask R-CNN only predicts a $28 \times 28$ segmentation mask inside each RoI, which is too coarse to capture the fine details of many objects. Methods such as

PointRend (Kirillov et al., 2020), RefineMask (Zhang et al., 2021a) and Mask Transfiner (Ke et al., 2022) therefore instead predict a $112 \times 112$ mask within each RoI, allowing for fine-grained segmentation predictions. PointRend achieves this by using an MLP, RefineMask by iteratively using their SFM module consisting of parallel convolutions with different dilations, and Mask Transfiner by using a transformer. However, all of these methods have limitations. PointRend has no access to neighboring features, RefineMask performs computation on all locations within the RoI at all times, and Mask Transfiner performs attention over *all* active features instead of over neighboring features only and it does not have access to passive features. EffSeg instead performs local computation at sparse locations while keeping access to both active and passive features.

Another family of methods obtaining fine-grained segmentation masks, are contour-based methods (Peng et al., 2020; Liu et al., 2021; Zhu et al., 2022). Contour-based methods first fit a polygon around an initial mask prediction, and then iteratively update the polygon vertices to improve the segmentation mask. Contour-based methods can hence be seen as a post-processing method to improve the quality of the initial mask. Contour-based methods obtain good improvements in mask quality when the initial mask is rather coarse (Zhu et al., 2022) (*e.g.* a mask predicted by Mask R-CNN (He et al., 2017)), but improvements are limited when the initial mask is already of high-quality (Zhu et al., 2022) (*e.g.* a mask predicted by RefineMask (Zhang et al., 2021a)).

**Spatial-wise dynamic networks.** In order to be efficient, EffSeg only performs processing at those spatial locations that are needed to obtain a fine-grained segmentation mask, avoiding unnecessary computation in the bulk of the object. EffSeg could hence be considered as a spatial-wise dynamic network. Spatial-wise dynamic networks have been used in many other computer vision tasks such as image classification (Verelst & Tuytelaars, 2020), object detection (Yang et al., 2022) and video recognition (Wang et al., 2022). However, these methods differ from EffSeg, as they apply an operation at sparse locations on a dense tensor (see SparseOnDense method from Sec. 3.2), whereas EffSeg uses the Structure-Preserving Sparsity (SPS) method separately storing the active features, the passive features, and a 2D index map containing the feature indices. This brings in two advantages: (1) passive features are not copied between subsequent sparse operations leading to increased storage efficiency, and (2) pointwise operations such as linear layers can directly be applied on the active features (instead of first having to select these from the 2D map) leading to increased processing speeds.

## 3 EffSeg

### 3.1 High-level overview

EffSeg is a two-stage instance segmentation head obtaining fine-grained segmentation masks by using a multi-stage refinement procedure similar to the one used in PointRend (Kirillov et al., 2020) and RefineMask (Zhang et al., 2021a). For each detected object, EffSeg first predicts a $14 \times 14$ mask within the RoI and iteratively upsamples this mask by a factor 2 to obtain a fine-grained $112 \times 112$ mask. See Figure 1 for an illustration of how EffSeg efficiently predicts fine-grained segmentation masks.

The $14 \times 14$ mask is computed by working on a dense 2D feature map of shape $[N_R, F_0, 14, 14]$, with $N_R$ the number of RoIs and $F_0$ the feature size at refinement stage 0. However, the $14 \times 14$ mask is too coarse to obtain accurate segmentation masks, as a single cell from the $14 \times 14$ grid might contain both object and background pixels, rendering a correct assignment impossible. To solve this issue, higher resolution masks are needed, reducing the fraction of ambiguous cells which contain both foreground and background.

The predicted $14 \times 14$ mask is therefore upsampled to a $28 \times 28$ mask, where in some locations the old predictions are overwritten by new ones, and where in the remaining locations the predictions are left unchanged. Features corresponding to the mask locations which require a new prediction, are called *active* features, whereas features corresponding to the remaining mask locations which are not updated, are called *passive* features. Given that a new segmentation prediction is only required for a subset of spatial locations within the $28 \times 28$ grid, it is inefficient to use a dense feature map of shape $[N_R, F_1, 28, 28]$ (as done in Refine-Mask (Zhang et al., 2021a)). Additionally, when upsampling by a factor 2, every grid cell gets subdivided in

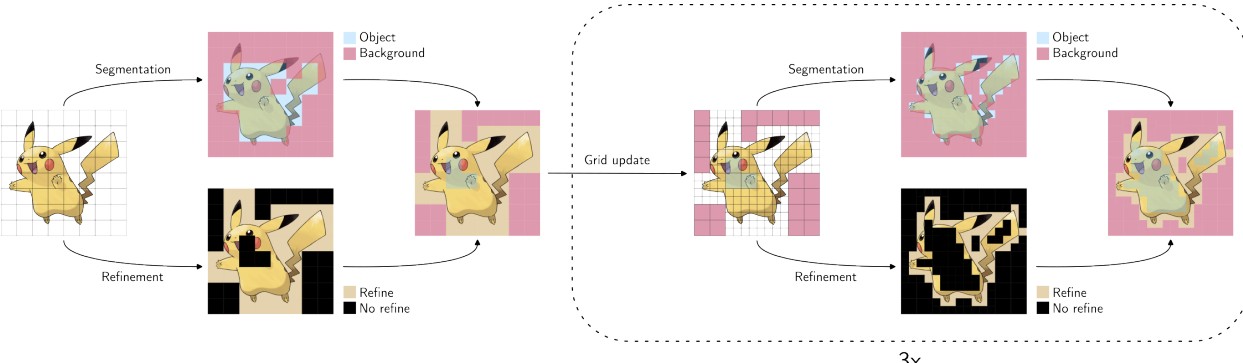

Figure 1: High-level overview of how EffSeg efficiently predicts fine-grained segmentation masks. EffSeg consists of multiple stages refining the predicted segmentation masks, with each stage consisting of a segmentation prediction (top branch) and a refinement prediction (bottom branch). By only refining (*i.e.* upsampling) at a select number of locations, EffSeg efficiently predicts fine-grained segmentations. In the example shown above, only 65% of the bounding box area needs to be processed in the second stage, and only roughly 35% and 20% in the third and fourth stages respectively.

a $2 \times 2$ grid of smaller cells, with the feature from the parent cell copied to the 4 children cells. The dense feature map of shape $[N_R, F_1, 28, 28]$ hence contains many duplicate features, which is a second source of inefficiency. EffSeg therefore introduces the Structure-Preserving Sparsity (SPS) method, which separately stores the active features, the passive features (without duplicates), and a 2D index map containing the feature indices (see Sec. 3.2 for more information).

EffSeg repeats this upsampling process two more times, resulting in the fine-grained $112 \times 112$ mask. Further upsampling the predicted mask is undesired, as $224 \times 224$ masks typically do not yield performance gains (Kirillov et al., 2020; Ke et al., 2022) while requiring additional computation. At last, the final segmentation mask is obtained by pasting the predicted $112 \times 112$ mask inside the corresponding RoI box using bilinear interpolation.

## 3.2 Structure-preserving sparsity

**Motivation.** When upsampling a segmentation mask by a factor 2, new predictions are only required in a subset of spatial locations. The **Dense** method, which consists of processing dense 2D feature maps as done in RefineMask (Zhang et al., 2021a), is inefficient as new predictions are computed over all spatial locations instead of only over the spatial locations of interest. A method capable of performing computation in a sparse set of 2D locations is therefore required. We distinguish following four sparse methods, where we first propose three baseline methods before introducing our SPS method.

First, the **Pointwise** method selects features from the desired spatial locations (called *active* features) and only processes these using pointwise networks such as MLPs or FFNs (Vaswani et al., 2017), as done in PointRend (Kirillov et al., 2020). Given that the pointwise networks do not require access to neighboring features, there is no need to store passive features, nor information about the 2D spatial relationship between features, making this method simple and efficient. However, the features solely processed by pointwise networks miss context information, resulting in inferior segmentation performance as empirically shown in Sec. 4.3. The Pointwise method is hence simple and efficient, but does not perform that well.

Second, the **Neighbors** method consists in both storing the active features, as well as their 8 neighboring features. This allows the active features to be processed by pointwise operations, as well as by non-dilated 2D convolutions with $3 \times 3$ kernel by accessing the neighboring features. The Neighbors method hence combines efficiency with access to the 8 neighboring features, yielding improved segmentation performance w.r.t. the Pointwise method. However, this approach is limited in the 2D operations it can perform. The 8 neighboring features for example do not suffice for 2D convolutions with kernels larger than $3 \times 3$ or dilated

Table 2: Comparison between dense and various sparse methods.

| Method | Example where used | Computationally efficient | Access to neighbors | Supports any 2D operation | Storage efficient |
|---|---|---|---|---|---|
| Dense | RefineMask (Zhang et al., 2021a) | ✗ | ✓ | ✓ | ✗ |
| Pointwise | PointRend (Kirillov et al., 2020) | ✓ | ✗ | ✗ | ✓ |
| Neighbors | - | ✓ | ✓ | ✗ | ✓ |
| SparseOnDense | - | ✓ | ✓ | ✓ | ✗ |
| SPS | EffSeg | ✓ | ✓ | ✓ | ✓ |

convolutions, nor do they suffice for 2D deformable convolutions which require features to be sampled from arbitrary locations. The Neighbors method hence lacks generality in the 2D operations it can perform.

Third, the **SparseOnDense** method consists in applying traditional operations such as 2D convolutions at sparse locations of a dense 2D feature map, as *e.g.* done in (Verelst & Tuytelaars, 2020). This method allows information to be exchanged between neighboring features (as opposed to the Pointwise method) and is compatible with any 2D operation (as opposed to the Neighbors method). Moreover, it is *computationally efficient* as it only performs computation there where it is needed. However, the use of a dense 2D feature map of shape $[N_R, F, H, W]$ as data structure is *storage inefficient*, given that only a subset of the dense 2D feature map gets updated each time, with unchanged features copied from one feature map to the other. Additionally, the dense 2D feature map also contains multiple duplicate features due to passive features covering multiple cells of the 2D grid, leading to a second source of storage inefficiency. Hence, while having good performance and while being computationally efficient, the SparseOnDense method is not storage efficient.

Fourth, the **Structure-Preserving Sparsity (SPS)** method stores a $N_A \times F$ matrix containing the active features, a $N_P \times F$ matrix containing the passive features (without duplicates) and a dense 2D index map of shape $[N_R, H, W]$ containing the feature indices. The goal of the index map is to *preserve* the 2D spatial configuration or *structure* of the features, such that any 2D operation can still be performed (as opposed to the Neighbors method). Separating the storage of active and passive features, enables SPS to update the active features without requiring to copy the unchanged passive features (as opposed to the SparseOnDense method). Moreover, by storing the active features in a dense $N_A \times F$ matrix, pointwise operations such linear layers can be applied without any data reorganization (as opposed to the SparseOnDense method), leading to increased processing speeds. The SPS method hence allows for fast and storage efficient sparse processing, while being computationally efficient and supporting any 2D operation thanks to the 2D index map.

An overview of the different methods with their properties is found in Tab. 2. The SPS method will be used in EffSeg as it ticks all the boxes.

**Toy example of SPS.** In Fig. 2, a toy example is shown illustrating how a non-dilated 2D convolution operation with $3 \times 3$ kernel is performed using the Structure-Preserving Sparsity (SPS) method. The example contains 4 active features and 3 passive features, organized in a $3 \times 3$ grid according to the dense 2D index map. Notice how the index map contains duplicate entries, with passive feature indices 5 and 6 appearing twice in the grid.

The SPS method applies the 2D convolution operation with $3 \times 3$ kernel and dilation 1 to each of the active features, by first gathering its neighboring features into a $3 \times 3$ grid, and then convolving this feature grid with the learned $3 \times 3$ convolution kernel. When a certain neighbor feature does not exist as it lies outside of the 2D index map, a padding feature is used instead. In practice, this padding feature corresponds to the zero vector.

As a result, each of the active features are sparsely updated by the 2D convolution operation, whereas the passive features and the dense 2D index map remain unchanged. Note that performing other types of 2D operations such as dilated or deformable (Dai et al., 2017) convolutions occurs in similar way, with the only difference being which neighboring features are gathered and how they are processed.

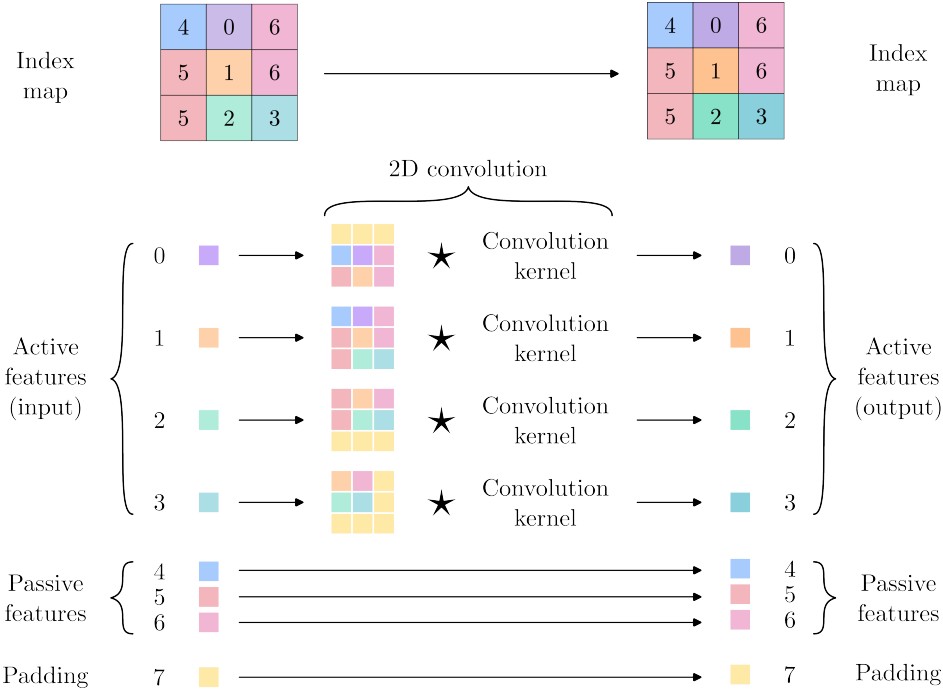

Figure 2: Toy example illustrating how a non-dilated 2D convolution operation with $3 \times 3$ kernel is performed using the SPS method. The colored squares represent the different feature vectors and the numbers correspond to the feature indices.

### 3.3 Detailed overview

Fig. 3 shows a detailed overview of the EffSeg architecture. The overall architecture is similar to the one used in RefineMask Zhang et al. (2021a), with some small tweaks as detailed below. In what follows, we provide more information about the various data structures and modules used in EffSeg.

**Inputs.** The inputs of EffSeg are the backbone feature maps, the predicted bounding boxes, and the query features. The backbone feature maps $B_s$ are feature maps coming from the $P_2$-$P_7$ backbone feature pyramid, with backbone feature map $B_s$ corresponding to refinement stage $s$. The initial backbone feature map $B_0$ is determined based on the size of the predicted bounding box, following the same scheme as in Mask R-CNN Lin et al. (2017); He et al. (2017) where $B_0 = P_{k_0}$ with

$$k_0 = 2 + \min\left(\lfloor \log_2(\sqrt{wh}/56) \rfloor, 3\right), \tag{1}$$

and with $w$ and $h$ the width and height of the predicted bounding box respectively. The backbone feature maps $B_s$ of later refinement stages use feature maps of twice the resolution compared to previous stage, unless no higher resolution feature map is available. In general, we hence have $B_s = P_{k_s}$ with

$$k_s = \max(k_0 - s, 2). \tag{2}$$

Note that this is different from RefineMask Zhang et al. (2021a), which uses $k_s = 2$ for stages 1, 2 and 3.

The remaining two inputs are the predicted bounding boxes and the query features, with one predicted bounding box and one query feature per detected object. The query feature is used by the detector to predict the class and bounding box of each detected object, and hence carries useful instance-level information condensed into a single feature.

**Dense processing.** The first refinement stage (*i.e.* stage 0) solely consists of dense processing on a 2D feature map.

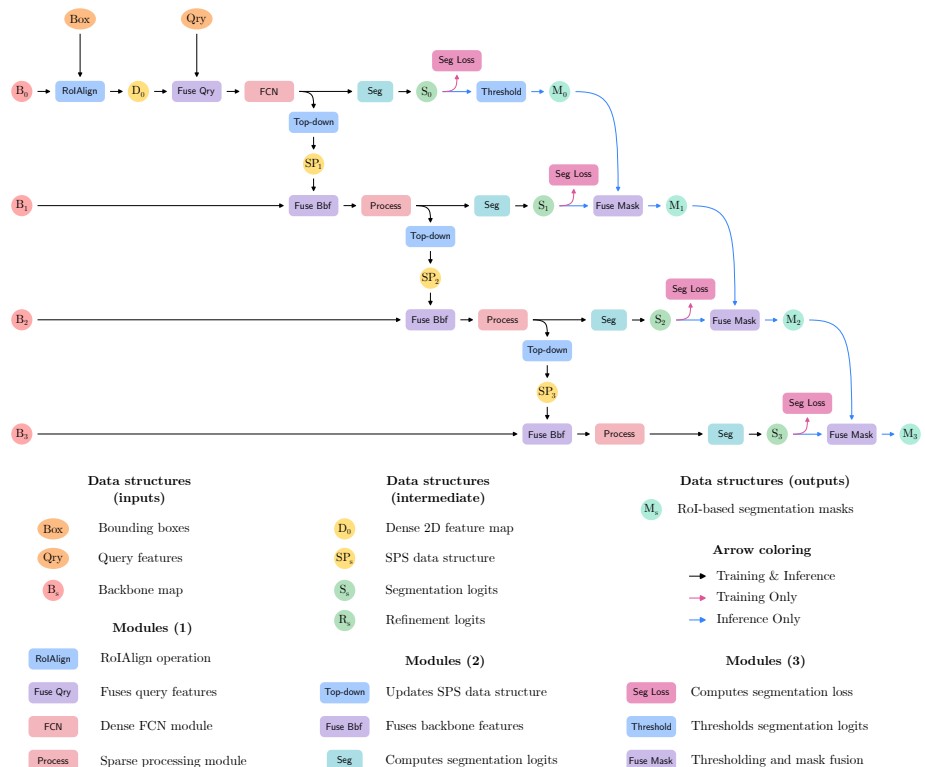

Figure 3: Detailed overview of the EffSeg architecture (the refinement branches and RoI mask pasting are omitted for clarity).

At first, EffSeg applies the RoIAlign operation He et al. (2017) on the $B_0$ backbone feature maps to obtain the initial RoI-based 2D feature map of shape $[N_R, F_0, H_0, W_0]$ with $N_R$ the number of RoIs (*i.e.* the number of detected objects), $F_0$ the feature size, $H_0$ the height of the map and $W_0$ the width of the map. Note that the numeral subscripts, as those found in $F_0$, $H_0$ and $W_0$, indicate the refinement stage. In practice, EffSeg uses $F_0 = 256$, $H_0 = 14$ and $W_0 = 14$.

Next, the query features from the detector are fused with the 2D feature map obtained by the RoIAlign operation. The fusion consists in concatenating each of the RoI features with their corresponding query feature, processing the concatenated features using a two-layer MLP, and adding the resulting features to the original RoI features. Fusing the query features allows to explicitly encode which object within the RoI box is considered the object of interest, as opposed to implicitly inferring this from the delineation of the RoI box. This is hence especially useful when having overlapping objects with similar bounding boxes.

After the query fusion, the 2D feature map gets further processed by a Fully Convolutional Network (FCN) Long et al. (2015), similar to the one used in Mask R-CNN He et al. (2017), consisting of 4 convolution layers separated by ReLU activations.

Finally, the resulting 2D feature map is used to obtain the coarse $14 \times 14$ segmentation predictions with a two-layer MLP. Additionally, EffSeg also uses a two-layer MLP to make refinement predictions, which are used to identify the cells (*i.e.* locations) from the $14 \times 14$ grid that require a higher resolution and hence need to be refined.

**Sparse processing.** The subsequent refinement stages (*i.e.* stages 1, 2 and 3) solely consist of sparse processing using the Structure-Preserving Sparsity (SPS) method (see Sec. 3.2 for more information about SPS).

At first, the SPS data structure is constructed or updated from previous stage. The $N_A$ features corresponding to the cells with the 10.000 highest refinement scores, are categorised as active features, whereas

the remaining $N_P$ features are labeled as passive features. The active and passive features are stored in $N_A \times F_{s-1}$ and $N_P \times F_{s-1}$ matrices respectively, with active feature indices ranging from 0 to $N_A - 1$ and with passive feature indices ranging from $N_A$ to $N_A + N_P - 1$. The dense 2D index map of the SPS data structure is constructed from the stage 0 dense 2D feature map or from the index map from previous stage, while taking the new feature indices into consideration due to the new split between active and passive features.

Thereafter, the SPS data structure is updated based on the upsampling of the feature grid by a factor 2. The number of active features $N_A$ increases by a factor 4, as each parent cell gets subdivided into 4 children cells. The children active features are computed from the parent active feature using a two-layer MLP, with a different MLP for each of the 4 children. The dense 2D index map is updated based on the new feature indices (as the number of active features increased) and by copying the feature indices from the parent cell of passive features to its children cells. Note that the passive features themselves remain unchanged.

Next, the active features are fused with their corresponding backbone feature, which is sampled from the backbone feature map $B_s$ in the center of the active feature cell. The fusion consists in concatenating each of the active features with their corresponding backbone feature, processing the concatenated features using a two-layer MLP, and adding the resulting features to the original active features.

Afterwards, the feature size of the active and passive features are divided by 2 using a shared one-layer MLP. We hence have $F_{s+1} = F_s/2$, decreasing the feature size by a factor 2 every refinement stage, as done in RefineMask Zhang et al. (2021a).

After decreasing the feature sizes, the active features are further updated using the processing module, which does most of the heavy computation. The processing module supports any 2D operation thanks to the versatility of the SPS method. Our default EffSeg implementation uses the Semantic Fusion Module (SFM) from RefineMask Zhang et al. (2021a), which fuses (*i.e.* adds) the features obtained by three parallel convolution layers using a $3 \times 3$ kernel and dilations 1, 3 and 5. In Sec. 4.3, we compare the performance of EffSeg heads using different processing modules.

Finally, the resulting active features are used to obtain the new segmentation and refinement predictions in their corresponding cells. Both the segmentation branch and the refinement branch use a two-layer MLP, as in stage 0.

**Training.** During training, EffSeg applies segmentation and refinement losses on the segmentation and refinement predictions from each EffSeg stage $s$, where each of these predictions are made for a particular cell from the 2D grid. The ground-truth segmentation targets are obtained by sampling the ground-truth mask in the center of the cell, and the ground-truth refinement targets are determined by evaluating whether the cell contains both foreground and background or not. We use the cross-entropy loss for both the segmentation and refinement losses, with loss weights $(0.25, 0.375, 0.375, 0.5)$ and $(0.25, 0.25, 0.25, 0.25)$ respectively for stages 0 to 3.

**Inference.** During inference, EffSeg additionally constructs the desired segmentation masks based on the segmentation predictions from each stage. The segmentation predictions from stage 0 already correspond to dense $14 \times 14$ segmentation masks, and hence do not require any post-processing. In each subsequent stage, the segmentation masks from previous stage are upsampled by a factor 2, and the sparse segmentation predictions are used to overwrite the old segmentation predictions in their corresponding cells. After performing this process for three refinement stages, the coarse $14 \times 14$ masks are upsampled to fine-grained $112 \times 112$ segmentation masks. Finally, the image-size segmentation masks are obtained by pasting the RoI-based $112 \times 112$ segmentation masks inside their corresponding RoI boxes using bilinear interpolation.

The segmentation confidence scores $s_{\mathrm{seg}}$ are computed by taking the product of the classification score $s_{\mathrm{cls}}$ and the mask score $s_{\mathrm{mask}}$ averaged over the predicted foreground pixels, which gives

$$s_{\mathrm{seg}} = s_{\mathrm{cls}} \cdot \frac{1}{|\mathcal{F}|} \sum_{i}^{\mathcal{F}} s_{\mathrm{mask},i} \tag{3}$$

with $\mathcal{F}$ the set of all predicted foreground pixels.

## 4 Experiments

### 4.1 Experimental setup

**Datasets.** We perform experiments on the COCO (Lin et al., 2014) instance segmentation benchmark. We train on the 2017 training set and evaluate on the 2017 validation and test-dev sets.

**Experiment details.** During our experiments, we use a ResNet-50+FPN or ResNet50+DeformEncoder backbone (He et al., 2016; Lin et al., 2017; Zhu et al., 2020) with the FQDet detector (Picron et al., 2022). For the ResNet-50 network (He et al., 2016), we use ImageNet (Deng et al., 2009) pretrained weights provided by TorchVision (version 1) and freeze the stem, stage 1 and BatchNorm (Ioffe & Szegedy, 2015) layers (see (Radosavovic et al., 2020) for the used terminology). For the FPN network (Lin et al., 2017), we use the implementation provided by MMDetection (Chen et al., 2019b). The FPN network outputs a $P_2$-$P_7$ feature pyramid, with the extra $P_6$ and $P_7$ feature maps computed from the $P_5$ feature map using convolutions and the ReLU activation function. For the DeformEncoder (Zhu et al., 2020), we use the same settings as in Mask DINO (Li et al., 2022), except that we use an FFN hidden feature size of 1024 instead of 2048. For the FQDet detector, we use the default settings from (Picron et al., 2022).

In order to determine the active and passive feature locations, EffSeg uses a separate refinement branch parallel to the segmentation mask branch (see Fig. 1). Here, the refinement branch predicts whether or not a feature location or cell should be refined, where during training a feature cell is labeled as positive when it contains both object and background pixels, and negative otherwise. The feature locations with the 10.000 highest refinement scores become the active feature locations, and the remaining locations are labeled as passive feature locations. See Appendix A for more information and for additional EffSeg implementation details.

We train our models using the AdamW optimizer (Loshchilov & Hutter, 2017) with weight decay $10^{-4}$. We use an initial learning rate of $10^{-5}$ for the backbone parameters and for the linear projection modules computing the MSDA (Zhu et al., 2020) sampling offsets used in the DeformEncoder and FQDet networks. For the remaining model parameters, we use an initial learning rate of $10^{-4}$. Our models are trained and evaluated on 2 GPUs with batch size 1 each.

We perform experiments using a 12-epoch or a 24-epoch training schedule, while using the multi-scale data augmentation scheme from DETR (Carion et al., 2020). The 12-epoch schedule multiplies the learning rate by 0.1 after the 9th epoch, and the 24-epoch schedule multiplies the learning rate by 0.1 after the 18th and 22nd epochs.

**Evaluation metrics.** When evaluating a model, we consider both its performance metrics as well as its computational cost metrics.

For the performance metrics, we report the Average Precision (AP) metrics (Lin et al., 2014) on the validation set, as well as the validation AP using LVIS (Gupta et al., 2019) annotations $AP^*$, and the validation AP using LVIS annotations with the boundary IoU (Cheng et al., 2021) metric $AP^{B*}$. For the main experiments in Sec. 4.2, we additionally report the test-dev Average Precision $AP_{test}$.

The reason why we also evaluate using LVIS annotations, is because the LVIS segmentation masks are of higher quality compared to the original COCO segmentation masks. The $AP^*$ and the $AP^{B*}$ metrics hence enable us to better evaluate the fine-grained quality of the predicted segmentation masks.

For the computational cost metrics, we report the number of model parameters, the number of GFLOPs during inference and the inference FPS. The number of inference GFLOPs and the inference FPS are computed based on the average over the first 100 images of the validation set. We use the tool from Detectron2 (Wu et al., 2019) to count the number of FLOPs and the inference speeds are measured on an NVIDIA A100-SXM4-80GB GPU.

**Baselines.** Our baselines are Mask R-CNN (He et al., 2017), PointRend (Kirillov et al., 2020) and RefineMask (Zhang et al., 2021a). Mask R-CNN could be considered as the entry-level baseline without any

Table 3: Main experiment results on COCO (see Sec. 4.1 for more information about the setup).

| Backbone | Detector | Seg. head | Epochs | AP | $AP_{50}$ | $AP_{75}$ | $AP_S$ | $AP_M$ | $AP_L$ | $AP_{test}$ | $AP^*$ | $AP^{B*}$ | Params | GFLOPs | FPS |
|---|---|---|---|---|---|---|---|---|---|---|---|---|---|---|---|
| R50+FPN | FQDet | Mask R-CNN++ | 12 | 38.8 | 59.1 | 42.2 | 19.4 | 41.2 | 57.4 | 39.3 | 40.9 | 28.6 | 37.5 M | 235.4 | 14.4 |
| R50+FPN | FQDet | PointRend++ | 12 | 39.5 | 59.3 | 42.9 | 19.5 | 42.2 | 58.9 | 40.1 | 42.4 | 31.9 | 37.8 M | 302.9 | 10.2 |
| R50+FPN | FQDet | RefineMask++ | 12 | 40.0 | 59.4 | **43.7** | 20.0 | 42.2 | **60.0** | 40.5 | **43.1** | **32.5** | 41.2 M | 446.3 | 10.3 |
| R50+FPN | FQDet | EffSeg (ours) | 12 | **40.1** | **59.7** | 43.5 | **20.1** | **42.8** | 59.4 | **40.5** | 42.9 | 32.4 | 38.8 M | 245.4 | 11.3 |
| R50+FPN | FQDet | Mask R-CNN++ | 24 | 39.5 | 60.2 | 43.0 | 19.6 | 42.0 | 57.5 | 40.4 | 41.7 | 29.4 | 37.5 M | 234.7 | 14.4 |
| R50+FPN | FQDet | PointRend++ | 24 | 40.6 | 60.7 | 44.2 | 21.0 | 43.1 | 60.0 | 41.2 | 43.2 | 32.4 | 37.8 M | 302.2 | 10.3 |
| R50+FPN | FQDet | RefineMask++ | 24 | 40.8 | 60.7 | 44.2 | 20.5 | 43.2 | 60.6 | **41.7** | **44.0** | **33.3** | 41.2 M | 445.7 | 10.3 |
| R50+FPN | FQDet | EffSeg (ours) | 24 | **41.1** | **61.1** | **44.7** | 20.7 | **43.6** | **60.9** | 41.6 | 43.8 | 33.0 | 38.8 M | 244.5 | 11.3 |
| R50+DefEnc | FQDet | Mask R-CNN++ | 12 | 40.7 | 61.7 | 44.2 | 21.8 | 43.4 | 59.3 | 41.7 | 43.4 | 30.9 | 45.0 M | 321.8 | 11.3 |
| R50+DefEnc | FQDet | PointRend++ | 12 | 41.5 | 62.0 | 45.0 | 22.3 | 44.2 | 60.9 | 42.5 | 44.3 | 33.7 | 45.3 M | 387.4 | 8.7 |
| R50+DefEnc | FQDet | RefineMask++ | 12 | 42.0 | **62.3** | **45.8** | **23.0** | 44.6 | **61.5** | **42.7** | **45.1** | **34.6** | 48.7 M | 529.1 | 8.7 |
| R50+DefEnc | FQDet | EffSeg (ours) | 12 | **42.1** | **62.3** | **45.8** | 22.1 | **44.8** | **61.5** | 42.6 | 45.0 | 34.4 | 46.3 M | 332.6 | 9.4 |

enhancements towards fine-grained segmentation. PointRend and RefineMask on the other hand are two baselines with improvements towards fine-grained segmentation, with RefineMask our main baseline due to its superior performance. We use the implementations from MMDetection (Chen et al., 2019b) for both the Mask R-CNN and PointRend models, whereas for RefineMask we use the latest version from the official implementation (Zhang et al., 2021a).

In order to provide a fair comparison with EffSeg, we consider the enhanced versions of above baselines, called Mask R-CNN++, PointRend++ and RefineMask++. The enhanced versions additionally perform query fusion and mask-based score weighting as done in EffSeg (see Appendix A). For PointRend++, we moreover replace the coarse MLP-based head by the same FCN-based head as used in Mask R-CNN, yielding improved performance without significant changes in computational cost.

Note that Mask Transfiner (Ke et al., 2022) is not used as baseline, due to irregularities in the reported experimental results and in the experimental settings as discussed in (Zhang & Ke, 2022).

## 4.2 Main experiments

Tab. 3 contains the main experiment results on COCO. We make following observations.

**Performance.** Performance-wise, we can see that Mask R-CNN++ performs the worst, that Refine-Mask++ and EffSeg perform the best, and that PointRend++ performs somewhere in between. This is in line with the arguments presented earlier.

Mask R-CNN++ predicts a $28 \times 28$ mask per RoI, which is too coarse to capture the fine details of many objects. This is especially true for large objects, as can be seen from the significantly lower $AP_L$ values compared to the other segmentation heads.

PointRend++ performs better compared to Mask R-CNN++ by predicting a $112 \times 112$ mask, yielding significant gains in the boundary accuracy $AP^{B*}$. However, PointRend++ does not access neighboring features during the refinement process, resulting in lower segmentation performance compared RefineMask++ and Effseg, which both do leverage the context provided by neighboring features.

Finally, we can see that the segmentation performance of both RefineMask++ and EffSeg is very similar. There are some small differences with RefineMask++ typically having higher $AP^*$ and $AP^{B*}$ values, and EffSeg typically having higher validation AP values, but none of these differences are deemed significant.

**Efficiency.** In Tab. 3, we can find the computational cost metrics of the different models *as a whole, i.e.* containing both the computational costs originating from the segmentation head as well as those originating from the backbone and the detector. To provide a better comparison between the different segmentation heads, we also report the computational cost metrics of the segmentation heads *alone* in Tab. 4.

As expected, we can see that Mask R-CNN++ is computationally the cheapest, given that it only predicts a $28 \times 28$ mask instead of a $112 \times 112$ mask. From the three remaining heads, RefineMask++ is clearly the most

Table 4: Computational cost metrics of the segmentation heads alone. The relative metrics (three rightmost columns) are computed w.r.t. to RefineMask++.

| Seg. head | Params | GFLOPs | FPS | Params decrease | GFLOPs decrease | FPS gain |
|---|---|---|---|---|---|---|
| Mask R-CNN++ | 2.9M | 70.3 | 98.7 | 56% | 75% | 272% |
| PointRend++ | 3.2M | 137.8 | 26.5 | 52% | 51% | 0% |
| RefineMask++ | 6.6M | 281.2 | 26.5 | 0% | 0% | 0% |
| EffSeg | 4.2M | 80.3 | 34.3 | **36%** | **71%** | **29%** |

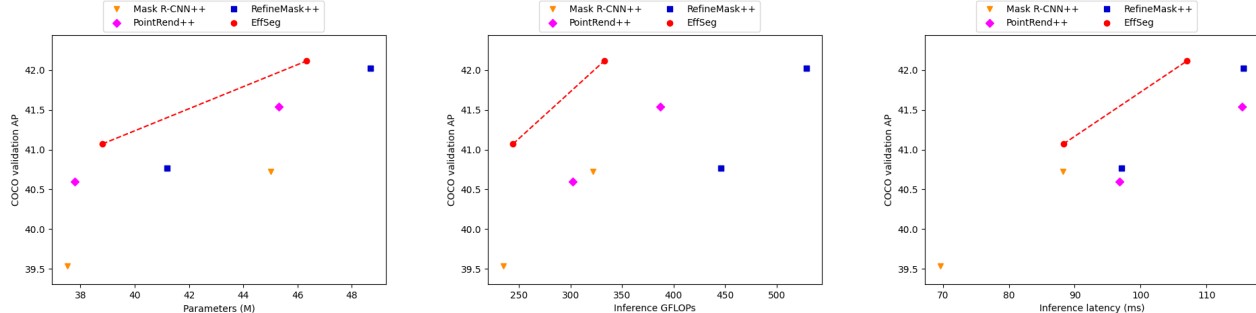

Figure 4: Performance vs. efficiency plots comparing the different segmentation models by plotting the COCO validation AP against the 'Parameters' (*left*), 'Inference GFLOPs' (*middle*) and the 'Inference FPS' (*right*) computational cost metrics.

expensive one, as it performs computation at all locations within the RoI instead of sparsely. PointRend++ and EffSeg are lying somewhere in between, being more expensive than Mask R-CNN++, but cheaper than RefineMask++.

Finally, when comparing RefineMask++ with EffSeg, we can see that EffSeg uses 36% fewer parameters, reduces the number of inference FLOPs by 71%, and increases the inference FPS by 29%.

**Performance vs. Efficiency.** Fig. 4 shows three performance vs. efficiency plots, comparing the COCO validation AP against the 'Parameters', 'Inference GFLOPs' and 'Inference FPS' computational cost metrics. From these, we can see that EffSeg provides the best performance vs. efficiency trade-off for each of the considered cost metrics.

We can hence conclude that EffSeg obtains excellent segmentation performance similar to RefineMask++ (*i.e.* the best performing baseline), while reducing the inference FLOPs by 71% and increasing the number of inference FPS by 29%.

### 4.3 Comparison between processing modules

In Tab. 5, we show results comparing EffSeg models with different processing modules (see Appendix A for more information about the processing module). All models were trained for 12 epochs using the ResNet-50+FPN backbone. We make following observations.

First, we can see that the MLP processing module performs the worst. This confirms that Pointwise networks such as MLPs yield sub-optimal segmentation performance due to their inability to access information from neighboring locations, as argued in Sec. 3.2.

Next, we consider the convolution (Conv), deformable convolution (Dai et al., 2017) (DeformConv) and Semantic Fusion Module (Zhang et al., 2021a) (SFM) processing modules. We can see that the Conv and DeformConv processing modules reach similar performance, whereas SFM obtains slightly higher segmentation performance. Note that the use of DeformConv and SFM processing modules was enabled by our SPS

Table 5: Comparison between different EffSeg processing modules on the 2017 COCO validation set.

| Seg. head | Module | AP | $AP_{50}$ | $AP_{75}$ | $AP_S$ | $AP_M$ | $AP_L$ | $AP^*$ | $AP^{B*}$ | Params | GFLOPs | FPS |
|---|---|---|---|---|---|---|---|---|---|---|---|---|
| EffSeg | MLP | 39.5 | 59.0 | 43.0 | 18.9 | 42.2 | 58.2 | 42.6 | 32.1 | 38.4 M | 227.3 | 12.2 |
| EffSeg | Conv | 39.8 | 59.4 | 43.1 | 19.4 | 42.0 | 59.3 | 42.6 | 32.0 | 38.5 M | 234.0 | 12.0 |
| EffSeg | DeformConv | 39.8 | 59.2 | 43.5 | 19.9 | 42.4 | 58.9 | 42.5 | 31.7 | 38.5 M | 235.0 | 11.5 |
| EffSeg | SFM | **40.1** | **59.7** | **43.5** | **20.1** | **42.8** | **59.4** | **42.9** | **32.4** | 38.8 M | 245.4 | 11.3 |
| EffSeg | Dense SFM | 39.8 | 59.1 | 43.5 | 19.5 | 42.5 | 59.0 | 42.8 | 32.3 | 38.9 M | 337.3 | 9.2 |

method (Sec. 3.2), which supports any 2D operation. This is in contrast to the Neighbors method (Sec. 3.2) for example, that neither supports DeformConv nor SFM (as it contains dilated convolutions). This hence highlights the importance of SPS to support any 2D operation, allowing for superior processing modules such as the SFM processing module.

Finally, Tab. 5 additionally contains the DenseSFM baseline, applying the SFM processing module over all RoI locations similar to RefineMask (Zhang et al., 2021a). When looking at the results, we can see that densely applying the SFM module (DenseSFM) as opposed to sparsely (SFM), does not yield any performance gains while dramatically increasing the computational cost. We hence conclude that no performance is sacrificed when performing sparse processing instead of dense processing.

### 4.4 Limitations and future work

The 2D operations (*e.g.* convolutions) performed on the SPS data structure, are currently implemented in a naive way using native PyTorch (Paszke et al., 2019) operations. Instead, these operations could be implemented in CUDA, which should result in additional speed-ups for our EffSeg models.

In EffSeg, a separate refinement branch is used to identify the spatial locations for which additional computation is performed (*i.e.* the active feature locations). In PointRend (Kirillov et al., 2020), the active feature locations are computed based on the segmentation mask uncertainties, and in RefineMask (Zhang et al., 2021a) the active feature locations are determined based on the boundaries of the predicted (and ground-truth) segmentation masks. Given these varying methodologies, it would be interesting to know which approach (if any) works best. Preliminary experiments seem to suggest that the different approaches reach similar performance, but further experimentation is required for a definite answer.

EffSeg can currently only be used for the instance segmentation task. Extending it to the more general panoptic segmentation (Kirillov et al., 2019) task, is left as future work.

## 5 Conclusion

In this work, we propose EffSeg performing fine-grained instance segmentation in an efficient way by introducing the Structure-Preserving Sparsity (SPS) method. SPS separately stores active features, passive features, and a dense 2D index map containing the feature indices, resulting in computational and storage-wise efficiency while supporting any 2D operation. EffSeg obtains similar segmentation performance as the highly competitive RefineMask head, while reducing the number of FLOPs by 71% and increasing the FPS by 29%.

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
