# OpenReview forum: "EffSeg: Efficient Fine-Grained Instance Segmentation using Structure-Preserving Sparsity"
_TMLR — Withdrawn by Authors_

### Review · Reviewer_Hwq6 · 2023-12-04

**Summary Of Contributions:**

This paper proposes EffSeg, a two-stage instance segmentation head that efficiently predicts fine-grained segmentation masks by using the Structure-Preserving Sparsity (SPS) method. SPS separately stores the active features, the passive features, and a 2D index map containing the feature indices, allowing for sparse computation while preserving the 2D structure of the features. EffSeg achieves similar performance to RefineMask while reducing the number of FLOPs by 71% and increasing the FPS by 29%. The paper also compares EffSeg with other baselines such as Mask R-CNN and PointRend, and shows the advantages of EffSeg in terms of performance and efficiency.

**Audience:**

Yes

**Claims And Evidence:**

Yes

**Requested Changes:**

**Paper Strength**

(1) The paper introduces an effective method for fine-grained instance segmentation, which is an important and challenging task in computer vision.
(2) The paper demonstrates the benefits of the SPS method, which enables fast and storage efficient sparse processing, while supporting any 2D operation and accessing neighboring features.
(3) The paper provides extensive experiments and ablation studies on the COCO dataset, using different backbones, detectors, and processing modules. The paper also evaluates the segmentation quality using LVIS annotations and the boundary IoU metric.

**Paper Weakness**

(1) The paper does not provide any qualitative results or visualizations of the predicted segmentation masks, which would help to better understand the performance and limitations of EffSeg.
(2) The paper does not compare EffSeg with any one-stage methods for instance segmentation, which are also relevant to the fine-grained segmentation task. As segmentation is a kind of clustering, some relevant papers could be included in this paper, e.g., Contrastive Clustering (aaai), Completer: Incomplete Multi-view Clustering via Contrastive Prediction (cvpr), Dual Contrastive Prediction for Incomplete Multi-View Representation Learning (pami), GroupViT: Semantic Segmentation Emerges from Text Supervision (cvpr).

(3) Given that the compared baselines, PointRend and RefineMask, were published two years ago, reviewing newly proposed baselines is encouraged to ensure up-to-date comparisons.
(4) How does EffSeg perform on other datasets or domains, such as PASCAL VOC or Cityscapes. Considering diverse object shapes, sizes, or levels of detail in different datasets could provide a more comprehensive evaluation.

**Strengths And Weaknesses:**

**Paper Strength**

(1) The paper introduces an effective method for fine-grained instance segmentation, which is an important and challenging task in computer vision.
(2) The paper demonstrates the benefits of the SPS method, which enables fast and storage efficient sparse processing, while supporting any 2D operation and accessing neighboring features.
(3) The paper provides extensive experiments and ablation studies on the COCO dataset, using different backbones, detectors, and processing modules. The paper also evaluates the segmentation quality using LVIS annotations and the boundary IoU metric.

**Paper Weakness**

(1) The paper does not provide any qualitative results or visualizations of the predicted segmentation masks, which would help to better understand the performance and limitations of EffSeg.
(2) The paper does not compare EffSeg with any one-stage methods for instance segmentation, which are also relevant to the fine-grained segmentation task. As segmentation is a kind of clustering, some relevant papers could be included in this paper, e.g., Contrastive Clustering (aaai), Completer: Incomplete Multi-view Clustering via Contrastive Prediction (cvpr), Dual Contrastive Prediction for Incomplete Multi-View Representation Learning (pami), GroupViT: Semantic Segmentation Emerges from Text Supervision (cvpr).

(3) Given that the compared baselines, PointRend and RefineMask, were published two years ago, reviewing newly proposed baselines is encouraged to ensure up-to-date comparisons.
(4) How does EffSeg perform on other datasets or domains, such as PASCAL VOC or Cityscapes. Considering diverse object shapes, sizes, or levels of detail in different datasets could provide a more comprehensive evaluation.

---

### Review · Reviewer_fcN4 · 2023-12-11

**Summary Of Contributions:**

This paper addresses the problem of image instance segmentation. The main contribution of this paper is the EffSeg architecture with the Structure Preserving Sparsity (SPS) method, which predicts object masks efficiently. EffSeg can be used with various two-stage instance segmentation models, leading to a faster execution speed without a performance drop.

**Audience:**

Yes

**Broader Impact Concerns:**

There are no significant broader impact concerns associated with this paper.

**Claims And Evidence:**

No

**Requested Changes:**

Several experiments could be added to enhance the paper:
- An experiment comparing the time complexity and storage complexity of SPS vs. the three baseline methods described in Sec 3.2, especially for SparseOnDense. This experiment is important, as it would clearly show the advantage of SPS.
- FLOPs are known to be better correlated with the execution speed on CPU than GPU. The authors could consider adding the CPU FPS, which should better show the advantage of EffSeg.

Also, please fix the typos if necessary (e.g., the FPS improvement numbers).

**Strengths And Weaknesses:**

Strengths:
- The paper is well-written and easy to read. For example, Figure 1 illustrates the idea well, and Figure 2 clearly describes the SPS method.
- The proposed method outperforms existing mask-prediction methods (i.e., segmentation heads) regarding the AP vs. computational complexity trade-off.

Weaknesses:
- The main weakness of the paper is its limited contributions. As the authors pointed out, the EffSeg architecture is similar to the one in RefineMask with minor tweaks. Also, the SPS method is an efficient way of implementing sparse operations on dense features, which is not a new architecture or algorithm. Table 3 also shows that EffSeg only marginally outperforms RefineMask++ regarding both AP and FPS.
- The authors claim that EffSeg achieves 29% higher FPS than RefineMask++ (in the abstract and Sec 4.2). However, this is not clear from Table 3. In the first block, EffSeg achieves 11.3 FPS, which is only 10% higher than RefineMask++.

---

### Review · Reviewer_fQMo · 2024-01-30

**Summary Of Contributions:**

EffSeg is a contribution to two-stage detector and segmentation research. It introduces a method called structure-preserving-sparsity (SPS) built off of prior works RefineMask and PointRend. SPS adopts a few smart considerations to efficiently store image features and allow for segmentation mask refinement when considering local features. Compared to recent two-stage CNN detectors, EffSeg reduces parameters, GFLOPS, and improves FPS for a negligible performance hit.

From all RoIs they cache features, with a lookup table for referencing the spatial arrangement as needed by local computations like convolutions. Features are either passive or active, where active features are those which need their mask refined further by local computation after upsampling and passive features are those which don’t (where a segmentation mask is not being updated, but from which boundary understanding might still be desired by the receptive field of a local convolution).

**Audience:**

Yes

**Claims And Evidence:**

Yes

**Requested Changes:**

The left and right side of Figure 2 are repetitive. The output active features have their color changed slightly to show their update, but not enough to be noticeable. This should be done more visibly.

The refinement branches and mask pasting should not be omitted from Figure 3. The figure has additional horizontal space and it would be helpful to include them even just as boxes with arrows. Also this figure is slightly repetitive, and could benefit from the same type of treatment as Figure 1 with a (3x) or something similar.

Ideally Figure 3 could also carry forward the Pikachu example from Figure 1 to highlight what features/parts of the grid were being processed

There are still a number of grammar issues.

**Strengths And Weaknesses:**

EffSeg improves over PointRend and RefineMask by being higher FPS, with fewer GFLOPS and comparable in performance. Although arguably more efficient, the authors should include more discussion earlier (rather than just briefly in limitations) about how custom operations that are supposed to be more efficient can actually be less efficient when they use high-level PyTorch code rather than custom CUDA operations. I would also expect that a work aiming to improve the efficiency of operations in two-stage detectors would contribute such custom code, as right now it is unverified whether the proposed changes actually do lead to speedup under a GPU/SIMD programming model. Given that the naive implementations seem to already be faster, this is omittable, but the paper would be served by this more thorough commitment to efficiency.

Most importantly, I believe that this paper ignores important related work that is worth consideration. The computer vision community is broad with lots of related work that makes it difficult to always include relevant comparisons.

In its most competitive setting with ResNet 50, EffSeg gets an AP of 42.1, running at 9.4 FPS, with 332 GFLOPS and 46.3M params. Mask DINO using a ResNet 50 gets an AP of 46.3, running at 14.2 FPS, with 286 GFLOPS and 52M parameters. Mask DINO has slightly more parameters but is both more accurate, faster, and uses less computation. MaskDINO also has a larger model that achieves MaskAP of 52.3. EVA achieves MaskAP of 55+. These related works are over a year old and the paper includes no mention of why comparisons aren’t made. Given that the research is focused on improving two-stage detectors, this paper has merit even if it doesn’t outperform related work. However, as it currently stands, the paper reads like there is not an additional body of related work that dramatically outperforms what's proposed.

Additionally, the authors dismiss a related work, Mask Transfiner, due to a GitHub issue thread. It is common practice to retrain a network under the same settings your own network was trained on when trying to make a valid comparison to prior work. This should have been done in the case of Mask Transfiner as it’s unclear for anyone who might want to use these methods how the two related works compare.

I would expect major revisions to include a comprehensive re-write that more adequately frames the research within the broader existing instance segmentation literature, as right now it reads as if PointRend or the “highly competitive RefineMask” are anywhere near top performance in instance segmentation. Furthermore, I would expect comparisons/mention to MaskDINO (which is cited but no further mention given – not sure if this is worse), under the clarity that there are considerable differences between the methodologies. I would also expect a Mask Transfiner to be re-trained appropriately to be a comparable baseline as it is clearly a highly related work. Overall, this paper is currently deceptive about the state of research in instance segmentation and by omitting mention of other methods does a disservice to a reader who may not realize the existing alternative approaches.

---

### Note · Authors · 2024-03-12

I have read and agree with the venue's withdrawal policy on behalf of myself and my co-authors.